# Development of Three-Axis Fibre-Optic Seismograph for Direct and Autonomous Monitoring of Rotational Events with Perspective of Historical Review

**DOI:** 10.3390/s22228902

**Published:** 2022-11-17

**Authors:** Anna T. Kurzych, Leszek R. Jaroszewicz, Jerzy K. Kowalski

**Affiliations:** 1Institute of Applied Physics, Military University of Technology, 00-908 Warsaw, Poland; 2Elproma Electronics Ltd., 05-152 Czosnów, Poland

**Keywords:** optical fibre sensor, rotational seismology, seismograph, detection, recording, earthquake

## Abstract

The paper presents historical perspective of fibre-optic seismographs designed and constructed at the Institute of Applied Physics at Military University of Technology, Poland based on the so-called minimum configuration of fibre-optic gyroscope. The briefly presented history, which originated in the 1998 by the system named GS-13P, laid solid foundations for the construction of a three-axis fibre-optic seismograph. The presented system meets all technical requirements of rotational seismology in terms of measurement parameters (measuring range from 10^−8^ rad/s to several rad/s and frequency from 0.01 Hz to 100 Hz) and utility features (mobility, autonomy, power independence, environmental stability). The presented device provides universal application both for research in engineering applications (high buildings, chimneys, wind towers) as well as in seismological research.

## 1. Introduction

Almost every week we hear information about strong earthquakes. Many of them are disastrous, causing thousands of deaths and devastating destruction [1,2,3,4]. Seismometers has been recording the motion of the ground during an earthquake for decades. They usually measure up and down motions and two horizontal motions [5]. However, the ground also rotates around three orthogonal axes and it has been very hard to record them [6,7]. Moreover, this has not been carried out routinely in seismology. Recent research [8,9,10,11,12,13] emphasizes a recording importance of six degrees of freedom of ground motion. The coupling of ground movement data of three rotational components with three translational ones can help event localization, earthquake source estimation, or wavefield separation. The installation of rotational sensors close to large seismic sources can create a unique opportunity to test some of these capabilities in a real-world application.

There are many different types of rotational sensors, such as mechanical (e.g., two antiparallel pendulum seismometers (TAPS) by Polish Academy of Science [14], Rotaphone by Czech Academy of Science [15]), MEMS technology (Horizon by EMCORE [16]), electromechanical (R1, R2 by Eentec [17]), and optical (Ring Laser gyroscope by LMU [18], optical fibre sensor blueSeis-3A by iXblue [19] or SRS-5000 by Optolink), which are widely comparted in [20]. However, the total insensitivity to linear motion, wide measuring range, high sensitivity, and portability make systems based on fibre-optic gyroscopes (FOG) the most appropriate sensors for the rotational seismology. The interferometric FOG exploiting quantum states of light are worth mentioning. Theoretically, they allow the fundamental limitation by the shot-noise limit to be overcome. The first attempts of such a system are not competitive with commercially available ones but they are a significant attempt to reach the ultimate sensitivity limits in the Sagnac interferometry [21].

It is now over 50 years since the era of optical fibre sensors started with the noncontact vibration monitoring sensor based on bifurcated fibre bundles (U.S.03327584; 1967) [22]. The first single-mode optical fibres appeared a decade later, which raised the possibility of designing interferometers, which promised immense engineering benefits compares to their free-space precursors bolted on optical tables. Optical fibre sensors offer a number of advantages (i.a., immunity to electromagnetic interference, minimal invasiveness, and light weight), making them a leading technology in many areas. The FOG based on the Sagnac interferometer is today arguably the most successful fibre sensor technology. The Sagnac effect makes it possible to directly obtain information about the component of the angular movement perpendicular to the sensor plane. Its principle was determined in 1913 [23], and its first fibre implementation appeared in 1976 by Vali and Shorthill as an operational FOG [24].

The paper emphasizes the importance of rotational seismology development with a brief description of the author’s experience in this field associated with rotational sensor construction, as well as a new three-axial system for the study of rotational movements in seismology as well as in engineering and technical sciences.

## 2. Systems Constructed at the Institute of Applied Physics at Military University of Technology, Poland

The exciting adventure with a system for angular velocity recording at the Institute of Applied Physics at the Military University of Technology, Poland, originated with the system named GS-13P in the 1990s [25] (Table 1). 

The system was characterized by the sensitivity equalling 3.49 × 10^−3^ rad/s. It included an optical head constructed according to the minimum FOG configuration using Hi-Bi fibre optic elements, a signal proceeding system based on EG&G lock-in 7260 type, and an IBM PC 5150 (IBM, Armonk, NY, USA) with special software PC-DOS. The next device, constructed in 2001, was named FORS-I (Fibre-Optic Rotational Seismometer) [26], with sensitivity equal to 2.2 × 10^−6^ rad/s. It used 400 m of Panda fibre. Subsequently, FORS-II was constructed (2003) [27] with sensitivity equal to 4.2 × 10^−8^ rad/s applying 11,000 m of single-mode optical fibre (SMF). FORS-I and -II have been successfully applied to record seismic rotational events in seismological observatories in Ojców and Książ, Poland. The next generation of FORS was a more advanced device with remote-control AFORS (Autonomous Fibre-Optic Rotational Seismograph) (2010) [28], which was constructed with a sensor loop of 15,000 m length of SMF and sensitivity at the level of 4.0 × 10^−9^ rad/s. AFORS effectively recorded the rotational effect in the period of 2010–2017 at the seismological observatory in Książ, Poland. In order to perform mobile, a remote device with wider measuring range, the FOSREM (Fibre-Optic System for Rotational Events and Phenomena Monitoring), was proposed in 2015. This rotational seismograph uses optimized 5 000 m of SMF and has sensitivity at level 2 × 10^−8^ rad/s. It meets almost all technical [20] requirements of the rotation seismology, both in seismological and engineering applications. Finally, based on FOSREM parameter optimization and its remote control, the sensors named FOS5 (Fibre-Optic Seismograph) were constructed and applied for the rotational seismology investigation. They were applied for rotational movement monitoring in buildings [29], in an international comparative sensor test [7], and for seismic event recording [14]. The area of FOS applications, especially as a three-axial system, is enormous (Figure 1), from seismic rotational wave observation in seismological observatories to wind farms or glory monitoring.

Having such instrumentation, it is extremely important to gather data and their analysis in order to better understand the origin of earthquakes, and in particular, to relate them to the geological context as well as to analyse engineering aspects of high and complex construction. Authors should discuss the results and how they can be interpreted from the perspective of previous studies and of the working hypotheses. The findings and their implications should be discussed in the broader context possible. Recently, our works are focused on the construction of a three-axis fibre-optic seismograph (FOS6) in order to obtain full rotational event data of ground motion. In comparison, the previous device, FOS6, applies a 6 km length of SMF. This value assures efficient sensitivity, which is analysed in paragraph 3, as well as the device’s mobility. The optical heads’ work has been improved by the amplified spontaneous emission (ASE) light source application in order to overcome the poor spectrum, polarization issues, and thermal instability of the applied previous superluminescent diodes. Additional application of the optical fibre circulator instead of the 2 × 2 coupler protects the better power budget. Moreover, FOS6 is synchronized with the UTC scale through the PTP protocol (precision time protocol), which ensures the synchronization of measurements of all autonomous seismographs with an accuracy of about 250 ns, regardless of the location on the globe. The time source is a time server located in the PCU (power communication unit) and collects the time via GNSS.

## 3. Construction of Three-Axis Fibre-Optic Seismograph—FOS6

All sensors that are part of the FOS6 based on the Sagnac fibre optic interferometer, which detects differences of travel time between the counter-propagating beams on the closed paths. The phase shift *φ_s_*, caused by the rotational component of motion Ω perpendicular to the closed fibre loop, can be expressed by [24]:(1)$$\varphi_{S}=\frac{2\pi LD}{\lambda c}\Omega$$
where *L* and *D* are the length and diameter of the fibre coil, respectively; *λ* is the wavelength of the light source, and *c* is the light speed in a vacuum. In order to meet the rotational seismology requirements, three 6 km long SMF coils are utilized and installed orthogonally in FOS6, as is shown in Figure 2. FOS6 is divided into two parts. The first one is the optical part and it is constructed according to the minimum gyro configuration. The second one is the electronic part which realizes a digital closed-loop signal processing approach [30].

An amplified spontaneous emission (ASE) light source has been applied in order to overcome the poor spectrum stability, polarization, and thermal instability of superluminescent diodes and to obtain a high-performance system. ASE provides two basic advantages: it emits unpolarized light, which reduces polarization nonreciprocities; the output power is high and can be coupled to the SMF. The emitted light by ASE is easily divided for three sensor loops by an optical fibre coupler [30] (Figure 2a). Due to ASE’s narrower spectrum in comparison to the superluminescent diode, the relative intensity noise (RIN) needs to be taken into consideration and reduced by the proper modulation techniques. The input power is split by the optical fibre coupler to the particular axes, and then guided to the fibre circulator, which assures a better power budget than the optical fibre coupler, like in the previous constructed sensors. The light modulation according to the closed-loop approach is realized by the multifunctional integrated optical chip (MIOC), which additionally divides the beam to the two counterpropagating waves in the sensor loop. The interfered light travels to an avalanche photodiode detector (APD), where the conversion of the optical signal to the electronic one takes place. The variable-power light beam is measured continuously by the sensor module. The electronic part proceeds and modulates the signal according to the digital phase ramp with steps and resets synchronized with the square-wave biasing modulation [31]. Moreover, the electronic part controls operation of the ASE module. An FPGA-based module coordinates the operation of the electronic part modules, controls algorithms, and communicates with external devices. The software is divided into modules and allows a single sensor loop to be controlled (Figure 3).

The optical signal that contains the angular rate data after conversion to the electrical signal is amplified by a transimpedance amplifier and an additional amplifier adapted to buffer of the ADC input. Samples are collected by the module that supports the ADC. Each sample then goes to the calculation module, where the closed-loop scheme controls the phase shift between the counter-propagating lightwaves. 

## 4. Calculation of the Theoretical Sensitivity of Particular Sensor Loop

Backscattered light from optical fibre defects, photon shot noise in the detector, thermal noise of electronical elements, and fluctuations in the light source intensity are the major sources of the random noise in FOG. Angle random walk (ARW) represents the random noise in the gyro signal output [32]. The application of the broadband light sources decreases effects of the light backscattering. The photon shot noise decreases proportionally to the square root of the power impinging the detector. Relative-intensity noise means noise of the optical-intensity light source, and at high-light-source powers becomes a dominant noise contributor. ARW can be theoretically determined by the following formula, which takes into consideration all of the main noise factors under the auxiliary modulation depth π/2 [33]: (2)$$ARW=\frac{\sqrt{2}\lambda c}{2\pi DL}\sqrt{\frac{4kT}{R\eta^{2}P^{2}}+\frac{ei_{d}}{\eta^{2}P^{2}}+\frac{e}{\eta P}+\frac{\lambda^{2}}{4c\Delta \lambda }}$$
where *λ—*central wavelength, *c—*speed of light, *D—*loop diameter, *L—*loop length (6000 m), *k*—Boltzmann’s constant, *T—*temperature (293 K), *R—*resistance of the transimpedance transducer of the photodetector device (20 kΩ), *η—*efficiency ratio of photodiode (0.85 A/W), *P—*incident lightwave power on the photodiode, *e—*elementary charge, *i_d_—*photodiode dark current (80 nA), Δ*λ—*spectral width of the light source (40 nm). According to Equation (2), the minimum sensitivity is influenced, in turn, by components under the root: thermal noise of the APD preamplifier, dark current of the detector, quantum noise, and excess noise of the source. Nevertheless, in case of the low power values on the detectors, e.g., equalling 100 μW, quantum noise is dominant, while the RIN is neglected [34]:(3)$$ARW=\frac{\sqrt{2}\lambda c}{2\pi DL}\sqrt{\frac{e}{\eta P\Delta t}}$$
where Δ*t* is the average integration time (for ARW = 1 s). The ARW defined by Equations (2) and (3) was calculated for the particular constructed optical part of FOS6 (Table 2). As can be seen, Equation (2) in relation to (3) states that for low values of the signal on the detector, only quantum noise is important for the ARW defined in this way. The calculated sensitivity values for each FOS6 axis are efficient for the seismological application of the rotational seismology.

## 5. Conclusions

Rotational seismology is one of the most developing fields of study. We are continuously interested in the process of providing appropriate rotational sensors. In this paper, we reviewed the systems constructed at the Institute of Applied Physics at the Military University of Technology, Poland. Over twenty years of experience gives our team a solid foundation to design a three-axis fibre-optic seismograph that completely meets the rotational seismology requirements for field application. The parameters of three constructed optical parts have been carefully selected in order to achieve the theoretical sensitivity of the order of few nrad/s. Results obtained in the present work provide new insight into the sensor loop as the heart of the three-axis fibre-optic seismograph. We believe that FOS6 is a promising tool for practically designing and analysing rotational seismology networks and will gather significant data in future.

## Figures and Tables

**Figure 1 sensors-22-08902-f001:**
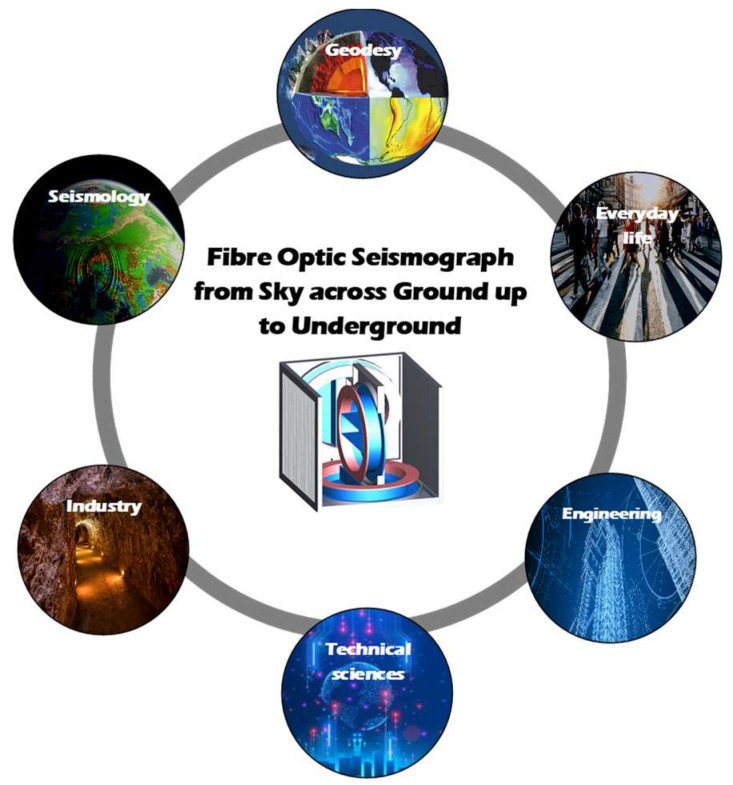
Visualization of three-axial fibre-optic seismograph with its wide range of applications.

**Figure 2 sensors-22-08902-f002:**
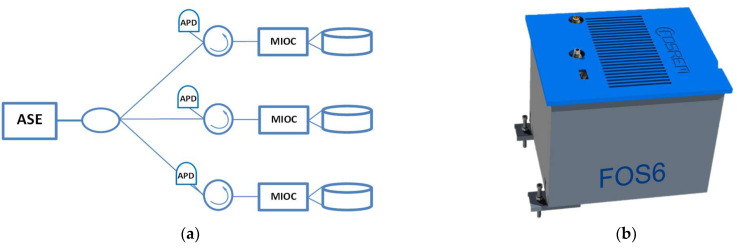
The basic schema of FOS6: (**a**) optical part, (**b**) system’s housing.

**Figure 3 sensors-22-08902-f003:**
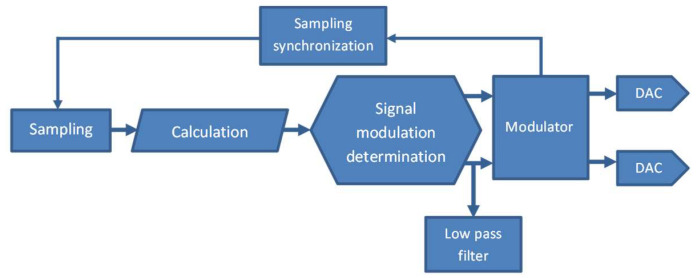
General structure of FOS6’s software.

**Table 1 sensors-22-08902-t001:** Historical brief of the fibre-optic seismograph constructed at the Institute of Applied Physics at the Military University of Technology, Poland.

Years	Name of the System	Parameters	Picture
1998	GS-13P	Ω_min_: 3.49 × 10^−3^ rad/s SL: 380 m Hi-Bi fibre, Radius: 0.1 m	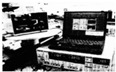
2001	FORS-I	Ω_min_: 2.2 × 10^−6^ rad/s Ω_max_: 4.8 × 10^−4^ rad/s SL: 400 m PANDA, Radius: 0.1 m	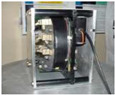
2004–2010	FORS-II (FOS1), AFORS (FOS2)	FORS-II: Ω_min_: 4.2 × 10^−8^ rad/s Ω_max_: 4.8 × 10^−4^ rad/s; SL: 11,000 m SMF Radius: 0.34 m; AFORS: Ω_min_: 4 × 10^−9^ rad/s, Ω_max_: 6.4∙10^−3^ rad/s SL: 15,000 m SMF, Radius: 0.34 m	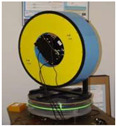
2015	FOSREM (FOS3 & FOS4)	Ω_min_: 2 × 10^−8^ rad/s, Ω_max_: few rad/s SL: 5000 m SMF, Radius: 0.125 m	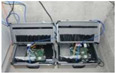
2018	FOS5	Ω_min_: 7 × 10^−8^ rad/s, Ω_max_: 10 rad/s SL: 5000 m SMF, Radius: 0.125 m	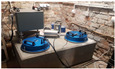

Ω_min_—min. detectable rotation rate, Ω_max_—max. detectable rotation rate, SL—length of the sensor loop, SMF—single-mode fibre.

**Table 2 sensors-22-08902-t002:** Theoretical ARW for the particular constructed optical part comprising the FOS6.

Axis	Length of the Optical Fibre [m]	Coil Diameter [m]	Coil Losses [dB]	Total Optical Losses [dB]	ARW Equation (2) [nrad/s]	ARW Equation (3) [nrad/s]
X	6 009		2.044	17.52	2.25	2.21
Y	6 021	0.215	1.914	16.89	2.08	2.06
Z	6 084		1.941	16.68	2.03	2.01

## Data Availability

Seismic events in miniSEED format registered last year by FOSREM sensors can be downloaded at: https://fosrem.eu/?page_id=462.

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
