# Peer review of "Development of Three-Axis Fibre-Optic Seismograph for Direct and Autonomous Monitoring of Rotational Events with Perspective of Historical Review"

_sensors, 2022, doi:10.3390/s22228902_

Round 1

Reviewer 1 Report

In the past 20 years, the authors have made outstanding achievements in the research of rotational seismometers. In this paper, the authors propose a novel 3-axis fibre-optic rotational seismograph, which may be a promising tool for rotational geology. I have the following suggestions.

1. Compared with previous versions of fiber-optic rotational seismometers, what performance of FOS6 is improved in this paper? A detailed comparison is required.

2. What key technical problems have been solved by FOS6 rotational seismometer? The authors need to make a detailed description.

3. Whether the software structure in Figure 3 is significantly different from the previous structure? If there is no difference, I don't think it is necessary  here.

Author Response

  1. Compared with previous versions of fiber-optic rotational seismometers, what performance of FOS6 is improved in this paper? A detailed comparison is required
  2. What key technical problems have been solved by FOS6 rotational seismometer? The authors need to make a detailed description.

The authors thank very much the Reviewers for the opinions which were very useful during the revision preparation. In the revised manuscript we have tried to improve our work according to the reviewers’ suggestions what is marked as underlined parts. Moreover, below we present our answers to all the questions with suitable comments. Finally, we believe that such improved paper will be acceptable to Reviewers for final publication.

The FOS6 gives possibility to measure three components of rotation. Because one optical source is used, the value of optical power needed to be higher and coupled to particular optical head by 1x3 coupler. Moreover, the time synchronization between the particular sensors – different seismographs is crucial. The paper gives also deeper analysis of the optical head sensitivity. The theoretical sensitivity has been calculated basing on two independent formula from various source [33], [34]. In comparison to the previous version of devices FOS6 applies also the optical fiber circulator instead of optical fiber coupler in order to get better power budget and ASE laser instead of SLED. The sensor loop parameter has been optimized for 6 km. Previously we applied 15 km and 5 km. The time synchronization between the particular optical part has been solved. We added the following statements in paragraph 2:

“In comparison the previous devices FOS6 applies 6 km length of SMF. This value assures efficient sensitivity, which is analyzed in paragraph 3, as well as device’s mobility. The optical heads work have been improved by the amplified spontaneous emission (ASE) light source application in order to overcome the poor spectrum, polarization issues and thermal instability of applied previous superluminescent diodes. Additional application of the optical fiber circulator instead of 2x2 coupler protects the better power budget. Moreover, FOS6 is synchronized with the UTC scale through the PTP protocol (Precision Time Protocol), which ensures synchronization of measurements of all autonomous seismographs with an accuracy of about 250 ns, regardless of the location on the globe. The time source is a time server located in the PCU (Power Communication Unit) and collecting the time via GNSS.”

  1. Whether the software structure in Figure 3 is significantly different from the previous structure? If there is no difference, I don't think it is necessary  here.

The software is different from the previous version because of the other type of electronics. Previously we used the open loop approach where the signal of two harmonics was measured or closed-loop but designed to 1-axis device. In the new version we applied the closed-loop approach with phase shift feedback loop compensation realized for 3-axes in one FPGA module. The software calculates the signal of modulation, so in our opinion it will be better to leave the schema in order to better understand the device operation.

All text has been checked regarding English improvements and typos elimination.

Reviewer 2 Report

The article may be accepted for publication provided that it is supplemented with earthquake records obtained using the described devices.

In the Introduction the sentence"Seismic  waves as sources of earthquakes..." contains a logical error.

Table 1 and the following paragraph completely duplicates each other.

The abbreviation RIN should be decoded (probably Relative Intensity Noise).

Author Response

The article may be accepted for publication provided that it is supplemented with earthquake records obtained using the described devices.

The authors thank very much the Reviewers for the opinions which were very useful during the revision preparation. In the revised manuscript we have tried to improve our work according to the reviewers’ suggestions what is marked as underlined parts. Moreover, below we present our answers to all the questions with suitable comments. Finally, we believe that such improved paper will be acceptable to Reviewers for final publication.

  1. In the Introduction the sentence "Seismic  waves as sources of earthquakes..." contains a logical error.

Excellent - Thank you for this comment! Yes, it is logical error. We changed the sentence to: Seismometers has been recording the motion of the ground during an earthquake for decades.

  1. Table 1 and the following paragraph completely duplicates each other.

We are agree with Reviewer opinion, but it is intentional description. Table 1 presents only synthesized main parameters of constructed devices. Whereas paragraph below Table 1 gives more precisely information about optical and electronic part construction as well as information about devices application in different places in nearly twenty years of their exploitation.

  1. The abbreviation RIN should be decoded (probably Relative Intensity Noise).

This abbreviation is decoded in the paragraph 2 under the Figure 2:

“An amplified spontaneous emission (ASE) light source has been applied in order to overcome the poor spectrum, polarization and thermal instability of superluminescent diodes and to obtain high-performance system. ASE provides two basic advantages: it emits unpolarized light which reduces polarization nonreciprocites; the output power is high and can be coupled to the SMF and using coupler easily shared for three sensors loops [30] [Figure 2(a)]. Due to ASE‘s narrower spectrum in comparison to superluminescent diode the relative intensity noise (RIN) needs to be taken into consideration and reduced by the proper modulation techniques.”

All text has been checked regarding English improvements and typos elimination.

Reviewer 3 Report

The manuscript under consideration is “Development of 3-Axis Fibre-Optic Seismograph for direct and autonomous monitoring of rotational events”. The paper historical review of fibre-optic seismograph designed in the Institute of Applied Physics at Military University of Technology.

From my point of view the manuscript is useful for scientists and practitioners in the field of seismology. There are some minor remarks.

1. Since this is a review it would be appropriate to mention this in the title, for example: “Development of 3-Axis Fibre-Optic Seismograph for direct and autonomous monitoring of rotational events: historical review”.

2. Kindly check numeration of paragraphs: number 2 is used twice with subsequent shift of numeration.

3. Kindly rename “Discussion” to “Conclusion”.

Round 2

Reviewer 2 Report

The revised version may be recommended for publication

Author Response

See revised version